# Comparison by Life-Cycle Assessment of Alternative Processes for Carvone and Verbenone Production

**DOI:** 10.3390/molecules27175479

**Published:** 2022-08-26

**Authors:** Jaime-Andrés Becerra, Juan-Miguel González, Aída-Luz Villa

**Affiliations:** Environmental Catalysis Research Group, Chemical Engineering Department, Engineering Faculty, Universidad de Antioquia UdeA, Calle 70 No. 52-21, Medellín 050010, Colombia

**Keywords:** carvone, catalyst, citrus, life cycle assessment (LCA), process design, turpentine, verbenone

## Abstract

Verbenone and carvone are allylic monoterpenoid ketones with many applications in the fine chemicals industry that can be obtained, respectively, from the allylic oxidation of *α*-pinene and limonene over a silica-supported iron hexadecachlorinated phthalocyanine (FePcCl_16_-NH_2_-SiO_2_) catalyst and with *t*-butyl hydroperoxide (TBHP) as oxidant. As there are no reported analyses of the environmental impacts associated with catalytic transformation of terpenes into value-added products that include the steps associated with synthesis of the catalyst and several options of raw materials in the process, this contribution reports the evaluation of the environmental impacts in the conceptual process to produce verbenone and carvone considering two scenarios (SI-raw-oils and SII-purified-oils). The impact categories were evaluated using ReCiPe and IPCC methods implemented in SimaPro 9.3 software. The environmental impacts in the synthesis of the heterogeneous catalyst FePcCl_16_-NH_2_-SiO_2_ showed that the highest burdens in terms of environmental impact come from the use of fossil fuel energy sources and solvents, which primarily affect human health. The most significant environmental impacts associated with carvone and verbenone production are global warming and fine particulate matter formation, with fewer environmental impacts associated with the process that starts directly from turpentine and orange oils (SI-raw-oils) instead of the previously extracted α-pinene and limonene (SII-purified-oils). As TBHP was identified as a hotspot in the production process of verbenone and carvone, it is necessary to choose a more environmentally friendly and energy-efficient oxidizing agent for the oxidation of turpentine and orange oils.

## 1. Introduction

Citrus essential oils extracted from citrus fruits peels (orange, mandarin, lemon, lime, and grapefruit) are principally constituted by *d*-limonene [1,2]. Turpentine oil extracted from pines is predominantly composed of *α*-pinene and *β*-pinene, among others [3,4]. Limonene, *α*-pinene, and *β*-pinene are used as solvents and precursors of fine chemicals [5,6,7]. Limonene and *α*-pinene are oxidized to produce carvone and verbenone (C_10_H_14_O), respectively; these allylic ketones are used in food, perfumes, medicine, agrochemicals, pesticides, and the chemical and oral hygiene product industries [8,9]. Although there is plenty of information about methods for producing the monoterpenoids verbenone and carvone at laboratory scale [8,10,11,12,13,14,15], hypothetical and industrially implemented technology processes for the production of these ketones are scarcely presented [16,17,18]. In addition, studies of, or approaches to the sustainability (environmental, economic, and social) assessment required to produce these compounds are not available. Since industrial verbenone and carvone production has been neither well known nor very detailed in the literature, the state of the art of industrial processes and the LCA for the production are still uncertain. The only studies concerning these topics have mainly focused on the evaluation of the environmental impacts of the extraction of some essential oils that contain terpenes as *α*-pinene or *d*-limonene.

It was found with the LCA methodology that in the evaluation of environmental sustainability of citrus plantations in China, the production and use of fertilizers are the first contributors to the environmental impacts [19]. Moura et al. [20] reported a study implementing the LCA methodology in the quantification of the environmental performance of lab scale essential oil production by hydro-distillation (HD), HD + lyophilization (HD + L), and supercritical fluid extraction (SFE) methods from the *Rosmarinus officinalis* L. species grown in Portugal. Their results showed that the energy is the main critical factor affecting the environmental performance of the extraction processes, particularly the lyophilization HD required to recover residual essential oil from the hydrolat. In addition, HD was preferable for *α*-pinene production from fresh and dried samples. Teigiserova et al. [21] implemented an LCA in a biorefinery for the production of limonene, citric acid, and animal feed from orange peel waste (OPW) generated from juice factories. Their results indicate that the climate change impact category, according to the Environmental Footprint life cycle impact assessment method, depended upon the electricity input with the highest CO_2,eq_ for current electricity mix, middle CO_2,eq_ using renewable sources, and the lowest CO_2,eq_ using electricity from wind. Santiago et al. [22] analyzed the environmental impacts with the LCA methodology in the valorization of citrus waste (CW) biorefineries targeting the production of *d*-limonene, among other co-products such as biogas and digestate. The scenarios covered four sections: pre-treatment, extraction (hydrodistillation, cold pressing, and solvent extraction), purification, and anaerobic digestion. The extraction and purification stages showed the main differences between the scenarios: the purification stage was primarily responsible for the highest environmental burdens in most of the scenarios due to the energy requirements; and in the scenario with the best profile, the pre-treatment and extraction steps were identified as hotspots due to the high electricity requirement. Jahandideh et al. [23] evaluated the environmental impacts with the LCA of a hypothetical limonene production facility using genetically engineered filamentous N_2_-fixing cyanobacteria. The analysis of the limonene production, as an alternative methodology for the manufacture of next-generation biofuels from renewable and sustainable sources, showed less negative environmental impact than lipid-based algal biodiesel production technologies. The major environmental burdens of the facility were the cyanobacteria nutrient supply sodium nitrate and the photobioreactor electrical requirements.

There are few reports on the environmental impacts related to the transformation of the main components of essential oils into high value-added compounds. Zhang et al. [24] reported on the environmental impacts in the production of sustainable biopolymers as an alternative for the conventional petroleum-based polymers, using the LCA methodology in the production of the biopolymer polylimonene carbonate through a conceptual process design from limonene oxidation with TBHP using different feedstocks (citrus waste and microalgae). The production stages for limonene oxide, TBHP, and the product were simulated in Aspen Plus V9 software to estimate both the energy consumption and mass balances for each process. Their results, obtained with the ReCiPe Mid-Point method, revealed that sustainable polylimonene carbonate synthesis was limited by the use of TBHP, suggesting that a more environmentally friendly and energy-efficient limonene oxidation method should be developed. In a previous report from our research group [25], we evaluated the environmental performance in the production of the fine chemical nopol, a monoterpenoid alcohol obtained via Prins reaction between *β*-pinene and paraformaldehyde through the implementation of the LCA methodology. The LCI was carried out combining primary and secondary data from simulation of the process using Aspen Plus software. The results obtained by the Hierarchist ReCiPe v1.13 (2008) method indicated that the extraction of raw materials contributed more to the environmental burdens than the production stages, and the solvent was identified as a hotspot. Some proposed alternatives to decrease the environmental burdens included the optimization of fossil resources, acting as solvent, and the evaluation of different heterogeneous catalysts and reaction conditions.

In this contribution, the conceptual design of the process for the production of verbenone and carvone by catalytic oxidation of limonene and α-pinene over the catalyst FePcCl_16_-NH_2_-SiO_2_ [26,27] is included together with the environmental performance through the implementation of the Life Cycle Assessment (LCA) methodology associated with these processes. To identify the effect of the feedstock on the environmental performance of the processes, two different alternatives in the proposed design of the process are analyzed according to the used feedstock: (i) raw commercial orange and turpentine oils (SI-raw-oils) or (ii) limonene and α-pinene obtained by a preliminary distillation of the essential oils (SII-purified-oils). The results from this research will contribute significantly to the state of the art of the implementation of LCA in the fine chemical processes of terpene-related compounds and the identification of the main environmental burdens which must be addressed to develop improvements in the processes.

## 2. Materials and Methods

### 2.1. General Introduction of the Development

This section includes a conceptual design of the reported [26,27] lab scale batch oxidation of limonene and *α*-pinene into carvone (~12% selectivity) and verbenone (~22% selectivity), respectively, using aqueous *t*-butyl hydroperoxide (TBHP) as an oxidizing agent and the heterogeneous catalyst FePcCl_16_-NH_2_-SiO_2_. Reactions are described in Figure 1. Two feedstocks are proposed for obtaining the allylic ketones either directly from the essential oils (SI-raw-oils) or from the terpenes that were previously extracted from the essential oils (SII-purified-oils). Then, the LCA methodology is used to identify the environmental impacts associated with the production of carvone and verbenone using the two feedstocks.

### 2.2. Description of the Heterogeneous Catalytic Process for the Batch Production of Verbenone and Carvone

The production of verbenone from the allylic oxidation of *α*-pinene (or turpentine oil) and the production of carvone from the allylic oxidation of *d*-limonene (or orange peel oil) were carried out through a proposed conceptual design of the scaled-up process shown in Figure 2 and with the design parameters presented in Table 1. The oxidation of the essential oils is promoted with TBHP as oxidant and over the iron phthalocyanine heterogeneous catalyst in an organic solvent. Two different process design alternatives or scenarios are proposed for the batch production of verbenone and carvone based on the purification of one of the raw materials. The first scenario (SI-raw-oils) uses raw commercial orange and turpentine oils to be directly processed in the reactor. The second scenario (SII-purified-oils) includes a preliminary distillation module for the purification of orange and turpentine essential oils. The process diagram was divided into the conventional raw material conditioning, reaction, and separation zones.

#### 2.2.1. Raw Materials Conditioning Zone

This zone includes the units required to carry out the conditioning of all the raw materials essential for the proposed processes. No further purification of the fresh input of solvent acetone is required since it is marketed with high purity. Although the use of dehydrated TBHP would be of great advantage, the purification of commercial aqueous solution is not considered in the process because of its unstable formulation resulting after higher purification [28]. The FePcCl_16_-NH_2_-SiO_2_ catalyst is obtained by the synthesis procedure reported in the literature [29,30]. Commercial turpentine and orange essential oils with no additional purification are used in the SI-raw-oils scenario of the process. However, the purification of *α*-pinene and *d*-limonene by fractional vacuum distillation in T-102 from the essential oils is proposed in the SII-purified-oils scenario based on a technical and economic feasibility analysis developed in previous studies [31].

#### 2.2.2. Reaction Zone

This zone contains the batch reactor R-101 in which the liquid-phase chemical transformation of the essential oils with the aqueous solution of the oxidizing agent TBHP takes place at constant temperature, using acetone as solvent and the heterogeneous catalyst FePcCl_16_-NH_2_-SiO_2_.

#### 2.2.3. Separation Zone

This zone contains the essential equipment to carry out the purification of verbenone and carvone. After the reaction takes place, the solid catalyst is separated from the reaction media using a filter F-101; the recovered catalyst may be reused at least seven times without losing significant catalytic activity [26,27]. The solvent acetone, TBA, and water are almost removed by simple distillation in D-101, and the acetone could be further purified by distillation if required according to the simulation results obtained in Aspen Plus software. The fractional distillation T-101 is used to purify the interest products verbenone and carvone under vacuum conditions.

### 2.3. LCA Methodology

The LCA was carried out following the ISO 14040 and 14044 standards [32,33] that describe the principles and framework for implementing the Life Cycle Assessment conducting the four phases: (i) Goal and Scope Definition, (ii) Life Cycle Inventory Analysis (LCI), (iii) Life Cycle Impact Assessment (LCIA), and (iv) Life Cycle Interpretation.

#### 2.3.1. Goal and Scope Definition

The goal of this study was to determine the environmental impacts in the two proposed alternatives or scenarios (SI-raw-oils and SII-purified-oils) of the conceptual design process for the batch production of verbenone and carvone, respectively, from turpentine and orange essential oil oxidation with TBHP and the heterogeneous FePcCl_16_-NH_2_-SiO_2_ catalyst. The functional unit chosen was the production of 1.0 kg/batch of carvone and 1.0 kg/batch of verbenone. The LCA was implemented using SimaPro 9.3.0.3 software by the cradle-to-gate approach, mass allocation, and system boundary, including the combination of five different stages shown in Figure 3. The stages of essential oil separation, reagent synthesis, and verbenone/carvone separation were supported with background data through the implementation of processes in databases and computational simulation models using Aspen Plus and Matlab software. The stages of the production of FePcCl_16_-NH_2_SiO_2_ and verbenone/carvone reaction were carried out at laboratory scale and conceptually scaled-up for the design of the process shown in Figure 2. In addition, the waste management of materials released in the process was not considered in the analysis. Two goals were identified for the present LCA; the first one is to determine the main causes of environmental impacts in the production of the heterogeneous iron base phthalocyanine catalyst FePcCl_16_-NH_2_SiO_2_, carvone, and verbenone. The second objective is to compare, in terms of environmental impacts, the production of verbenone and carvone, respectively, to the allylic oxidation of raw essential oils of turpentine and orange (SI-raw-oils), or the oxidation of their main components, *α*-pinene and *d*-limonene, previously purified from the essential oils (SII-purified-oils).

#### 2.3.2. Life Cycle Inventory Analysis

All data required for the LCI were obtained combining both secondary (background data) and primary (adjusted foreground data) sources (Figure 3). This is because this study is focused on an original conceptual process design to produce verbenone and carvone, and due to the absence of information on the implementation of the process at the industry level. The foreground data were obtained from laboratory scale experiments related to catalyst and product preparation in the reactor and adjusted to the scale of the process shown in Figure 2 by process simulation. The secondary data were obtained by computational simulation of mathematical models of the equipment in the process using Aspen Plus and Matlab software, from global LCI datasets such as Ecoinvent and Agrifootprint, and from patents and articles. For the implementation of the LCI in the SimaPro software, some additional processes were implemented since there was no available information in the selected databases (see Appendix A). The additional processes were based on either patented or reported information covering the production of orange essential oil [34] and the synthesis of several reagents such as *t*-butyl hydroperoxide [35,36], (3-aminopropyl)triethoxysilane [37], allylamine [38], ammonium heptamolybdate [39], tetrachlorophthalic anhydride [40], triethoxysilane [41], and FePcCl_16_-NH_2_-SiO_2_ catalyst by the chemical routes shown in the Appendix A. The available data of raw materials emission and extraction were obtained from the Agri-footprint 5.0—mass allocation and the Ecoinvent 3.8—allocation, cut-off by classification databases (Appendix A). The LCI is structured in the process block diagram shown in Figure 4 with a detailed description in the Appendix A.

#### 2.3.3. Life Cycle Impact Assessment

The LCIA of the process for the production of verbenone and carvone was implemented in SimaPro 9.3.0.3 software [42], using both ReCiPe 2016 Endpoint (H) v1.06/World (2010) H/A and IPCC 2021 GWP100 v1.0 methods. Additionally, results for the ReCiPe Midpoint are shown in the Appendix A and are reported to have lower uncertainty than the Endpoint method. However, the ReCiPe Endpoint provides better information on the environmental relevance of the environmental flows [43], and therefore an uncertainty analysis was considered for this method. The impact categories selected for the analysis in the ReCiPe method include global warming (human health, terrestrial ecosystems, and freshwater ecosystem), stratospheric ozone depletion, ionizing radiation, ozone formation (human health and terrestrial ecosystems), fine particulate matter formation, terrestrial acidification, eutrophication (fresh water and marine), ecotoxicity (fresh water, marine, and terrestrial), water toxicity, human toxicity (carcinogenic and non-carcinogenic), land use, resource scarcity (mineral and fossil), and water consumption (human health, terrestrial, and aquatic ecosystems). The IPCC 2021 GWP100 method was used to evaluate at a timeframe of 100 years the impacts of the synthesis of the materials on climate change, its implications, and future risks in terms of CO_2_ emissions including the GWP100 impact categories of fossil, biogenic, and land transformation [44].

## 3. Results and Discussion

### 3.1. Environmental Impacts in the Production of the Iron-Based Phthalocyanine Complex Catalyst FePcCl_16_-NH_2_-SiO_2_

The process contribution to the environmental impact categories (Table 2) by the ReCiPe method in the synthesis of the active phase complex FePcCl_16_ is presented in Figure 5. From the damage environmental impact categories perspective, the highest burdens are associated with potential damage to human health (Figure 5b), since the two highest impact categories using this synthesis procedure relate to the impact of global warming on human health followed by the fine particulate matter formation, and with minor weight for both carcinogenic and non-carcinogenic toxicity to humans (Figure 5a). Additionally, fossil resource scarcity and global warming in terrestrial ecosystems are, in a minor grade, important potential environmental impact categories to be considered from this process. In the impact categories, the process contributing the most to the impacts are the use of natural gas as a heating source, the requirement of organic solvents— mainly acetone and nitrobenzene—and the use of sulfuric acid, which is the highest contributing process in the human non-carcinogenic toxicity category.

When comparing the production of 1.0 kg of the modified support NH_2_-SiO_2_ and the catalyst with the active phase complex FePcCl_16_, similar results were observed in terms of process contribution and the most relevant impact categories (Figure 6 and Figure 7). However, it can be concluded that the synthesis of the support generates the highest environmental impacts between them—in this case, due to the large extent of inert atmosphere of Ar needed for the synthesis (Figure 6). LCA results using the IPCC GWP 100 method are used to estimate the 100-year global warming potential (GWP) from the catalyst synthesis. Like the results obtained through the ReCiPe method, it is shown that the synthesis of the support would generate more CO_2-eq_ than the active phase and the catalyst synthesis (Figure 7b).

### 3.2. Comparing Scenarios SI-Raw-Oils and SII-Purified-Oils in the Proposed Process to Produce Verbenone and Carvone

The main difference between SI-raw-oils and SII-purified-oils consists in the use of raw essential oil during the reaction step (turpentine or orange oil) in the first scenario and the inclusion of a distillation tower T-102 in the second scenario to purify the relevant monoterpene (*α*-pinene or *d*-limonene) prior to the reaction step. This causes different compositions of by-products in the process units of the SI-raw-oils and SII-purified-oils scenarios. The LCA results for both impact and damage environmental impact categories for these scenarios in the production of verbenone and carvone are presented in Figure 8 and Figure 9, respectively. The most relevant environmental impact categories in the systems are the fine particulate matter formation and the effect of global warming on human health, whereas the highest damage environmental impact categories are associated with human health issues. A comparison of SI-raw-oils and SII-purified-oils in terms of LCA results indicates that including a separation step in the process for purifying the raw essential oil increases the environmental impacts of producing verbenone without evident benefits in the production performance (Figure 8). In the case of carvone, the LCA results in Figure 9 show similar impacts for the two evaluated scenarios, which is due to the high concentration of *d*-limonene in the orange oil, causing low operating costs in the oil purification unit.

The contribution of the most important activities on the environmental impacts in the scenarios for the production of verbenone and carvone are presented in Figure 10 and Figure 11, respectively. Calculations of the impact on human health are presented since it was identified in the damage category results as the highest damage category contributing to the environmental impacts in the process, as shown in Figure 8b and Figure 9b. Liquified petroleum gas, transport trucks, and hydrochloric acid are the most significant contributors to human health impacts. Electricity is also a major impact process due to its intensive use in operating compressors and pumps, especially during the production of *t*-butyl hydroperoxide, and in refrigerating essential oils to avoid their degradation. In addition, high amounts of oxygen, hydrochloric acid, and butane (simulated in this study as liquefied petroleum gas) are needed to produce *t*-butyl hydroperoxide and heat used to operate distillation units. Acetone and the catalysts are not significant pollutants since they are constantly recirculated in the processes. However, *t*-butyl hydroperoxide is easily degraded to *t*-butanol, making its reuse unfeasible. In particular, the implementation of the orange processing step for the extraction of the orange oil in the carvone production increases the environmental impact of the process in comparison with the verbenone production (Figure 11). This is also evidenced from the results of the global warming potential calculated by the IPCC GWP100 method shown in Figure 12, where the amount of CO_2-eq_/kg carvone is almost 3.9 higher than the amount of CO_2-eq_/kg verbenone with respect to contaminant SII-purified-oils scenario.

An uncertainty analysis of the environmental impacts of the different scenarios of the process is shown in Figure 13. The single score probability results of the environmental impacts to produce 1 kg/batch of carvone were higher than the probabilities in the production of verbenone. However, the differences between the statistical parameters reported in Table 3 were very similar for the production of carvone since the orange essential oil has high levels of *d*-limonene concentration and the results do not affect both scenarios evaluated, as presented in Figure 9b and Figure 12b.

In this LCA, the production process of TBHP was identified as a hotspot in accordance with the network diagram of the process shown in Figure 14. Thus, a sensitive analysis was carried out considering the SI-raw-oils scenario, which has the lowest environmental impacts, to produce the allylic ketone and including the most important activities affecting the production of verbenone: transportation and type of electricity for TBHP production. Carvone production was not considered since the uncertainty analysis was very similar in both scenarios (coefficient of variations CV reported in Table 3 were very similar and lower than in the scenarios for verbenone production). EURO1, EURO3, and EURO5 were analyzed in the sensitive analysis of the transport by truck, and the electricity energy grid mix by hydroelectrical and natural gas sources as the most representative uses in Colombia were also considered in the results of the cumulative score for the input and outputs of the process (Table 4). The lowest cumulative scores in the production of TBHP (85.2%), i.e., the lowest environmental impact condition, was obtained using the EURO5 transport truck; however, the differences in using EURO1 and EURO3 were not significant. On the other hand, using electricity from natural gas (cumulative score 69.8%) rather than from hydroelectrical (cumulative score 51.2%) sources may have a greater contribution to the environmental impacts in the production of TBHP and therefore in the production of verbenone in the SI-raw-oils scenario.

## 4. Conclusions

As part of the conceptual design of a process to produce verbenone and carvone using turpentine and orange essential oil oxidation with TBHP and the heterogeneous catalyst FePcCl_16_-NH_2_-SiO_2_, an analysis of the environmental impacts was successfully applied with the LCA methodology. The evaluation of the environmental impacts in the synthesis of the heterogeneous catalyst FePcCl_16_-NH_2_-SiO_2_ showed that the greatest contributions to the environmental impacts come from the use of natural gas as an energy source and the requirement of both organic and inorganic solvents, which mainly affect human health. After performing a life cycle assessment to the verbenone and carvone processes of production through the SI-raw-oils and SII-purified-oils scenarios, it was concluded that the most important environmental impacts on human health are global warming and formation of fine particulate matter. When comparing SI and SII, it was observed that SI presented fewer environmental impacts, which means that it is a good alternative to use raw turpentine and orange oil to produce verbenone and carvone, respectively. Since orange oil has a high concentration of *d*-limonene (up 96 wt.%), environmental impact results were quite similar in both SI-raw-oils and SII-purified-oils scenarios, which did not happen in the case of verbenone production from turpentine oil, which has a lower concentration of α-pinene. TBHP was identified as a hotspot in the production of verbenone and carvone, thus suggesting that a more environmentally friendly and energy-efficient oxidizing agent method should be analyzed in the oxidation of turpentine and orange oils. The most important activities affecting the production of verbenone were transportation and type of electricity for TBHP production.

## Figures and Tables

**Figure 1 molecules-27-05479-f001:**
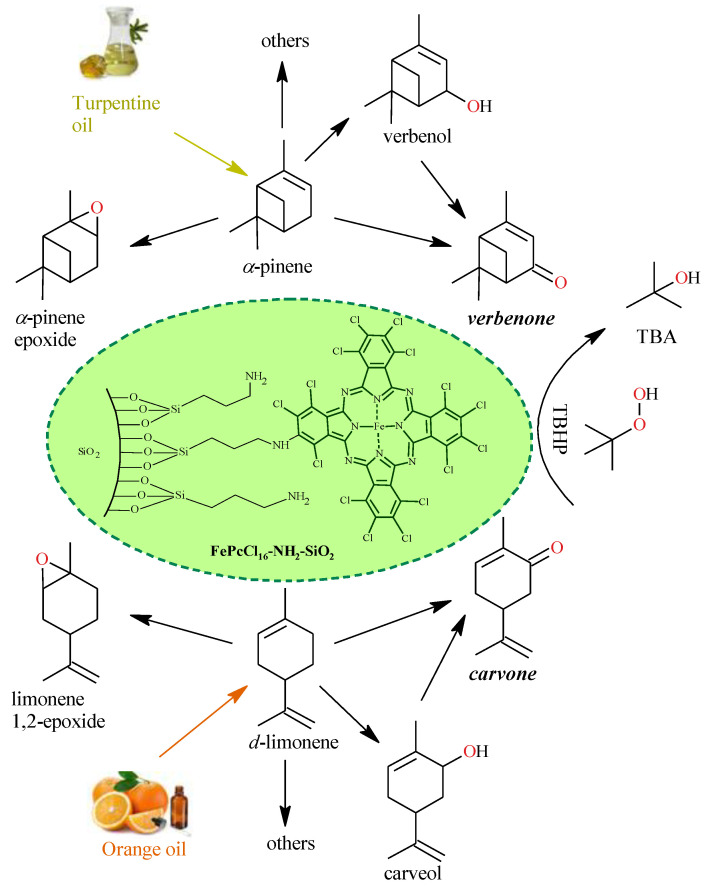
Chemical route for the production of verbenone and carvone from *α*-pinene and *d*-limonene oxidation with TBHP and FePcCl_16_-NH_2_-SiO_2_. Conditions: acetone, 40–50 °C, 6 h, molar ratio TBHP/monoterpene of 2.6:1. TBA: t-butyl alcohol.

**Figure 2 molecules-27-05479-f002:**
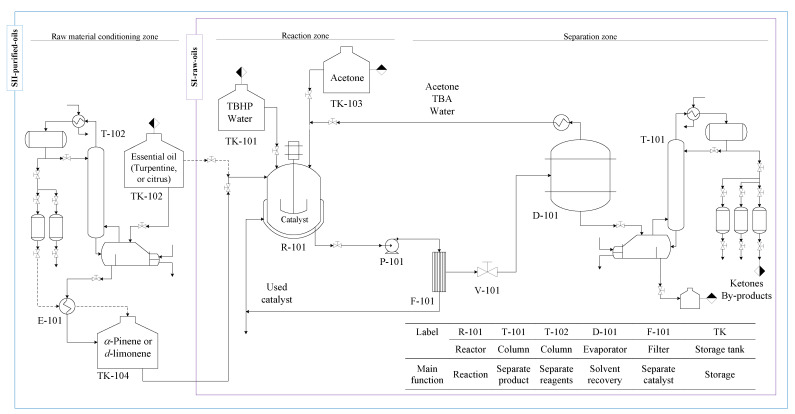
Process flow diagram for two scenarios (SI-raw-oils and SII-purified-oils) in the batch production of verbenone and carvone from turpentine and orange oils oxidation with TBHP and FePcCl_16_-NH_2_-SiO_2_.

**Figure 3 molecules-27-05479-f003:**
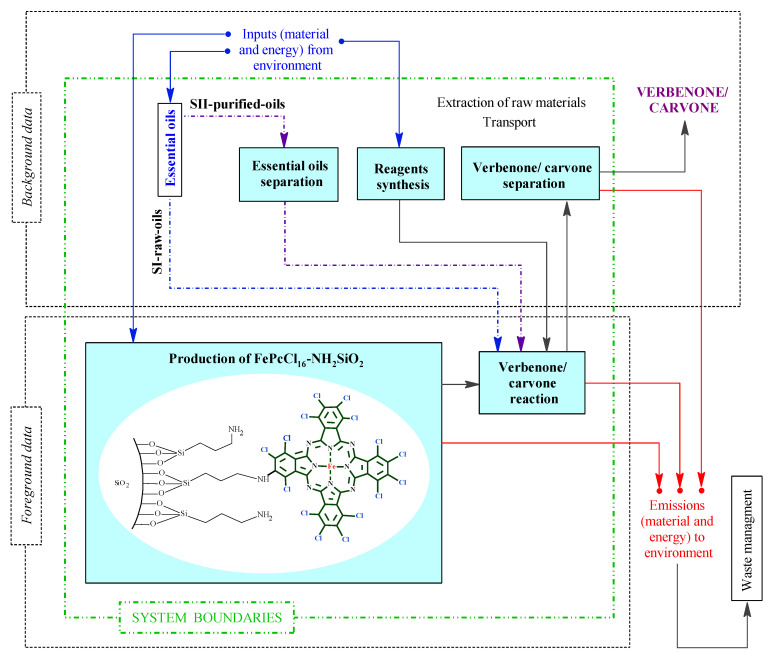
System boundaries corresponding to the scenarios in the LCA for verbenone and carvone production. Scenario I (SI-raw-oils), Scenario II (SII-purified-oils).

**Figure 4 molecules-27-05479-f004:**
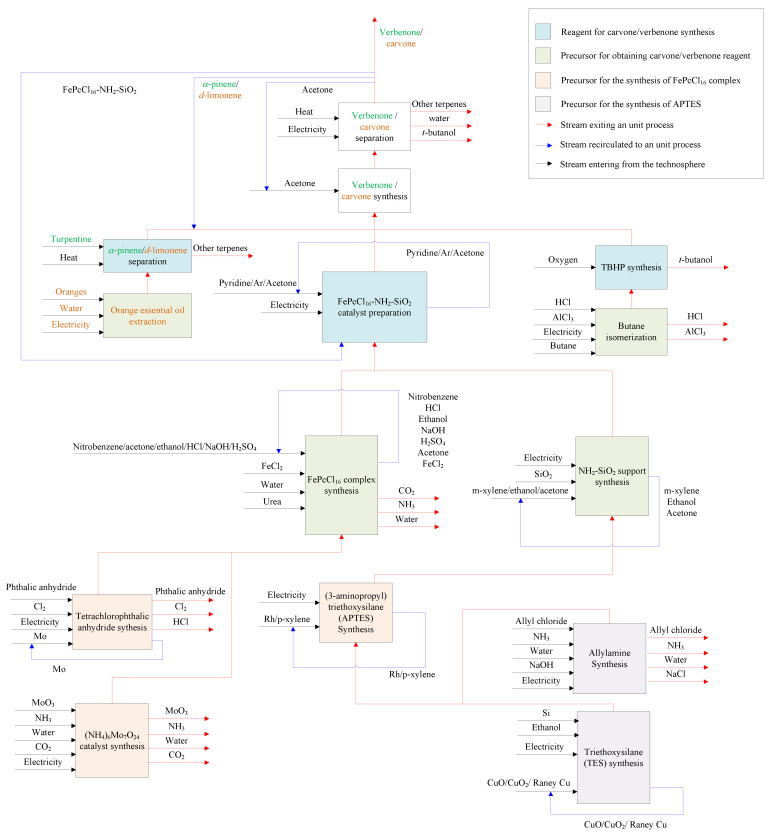
Process block diagram of the LCI for the batch production of verbenone and carvone from turpentine and orange oils oxidation with TBHP and FePcCl_16_-NH_2_-SiO_2_ (See detailed description in Appendix A).

**Figure 5 molecules-27-05479-f005:**
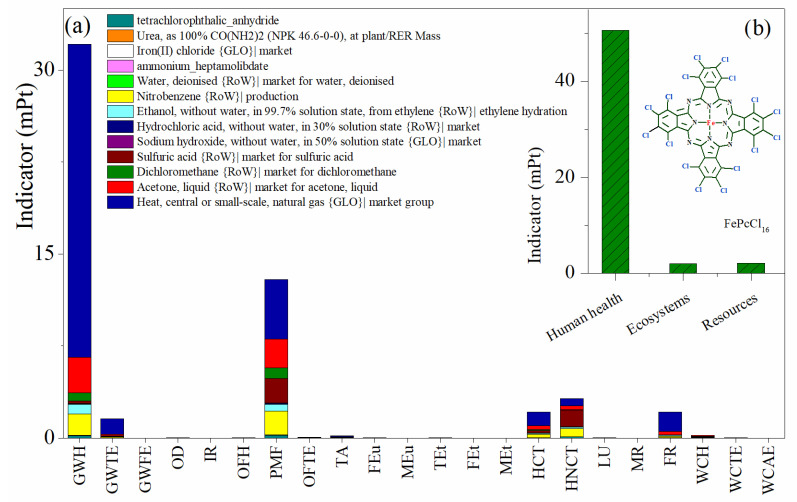
Environmental impact assessment by weighting indicators in the production of FePcCl_16_. (**a**) Impact categories by process contribution and (**b**) damage categories. Assessment method: ReCiPe 2016 Endpoint (H) V1.06/World (2010) H/A; mPt: milipoints.

**Figure 6 molecules-27-05479-f006:**
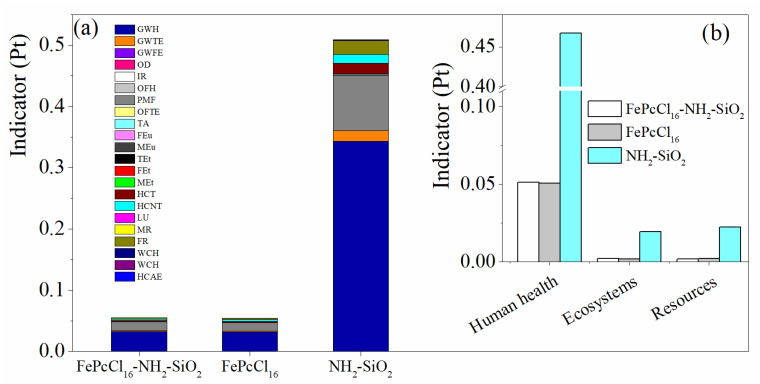
Comparison of the environmental impacts assessment by weighting indicators in the production of the catalyst FePcCl_16_-NH_2_-SiO_2_, support and active phase_._ (**a**) Impact categories by single score and (**b**) damage categories. Assessment method: ReCiPe 2016 Endpoint (H) V1.06/World (2010) H/A; Pt: points.

**Figure 7 molecules-27-05479-f007:**
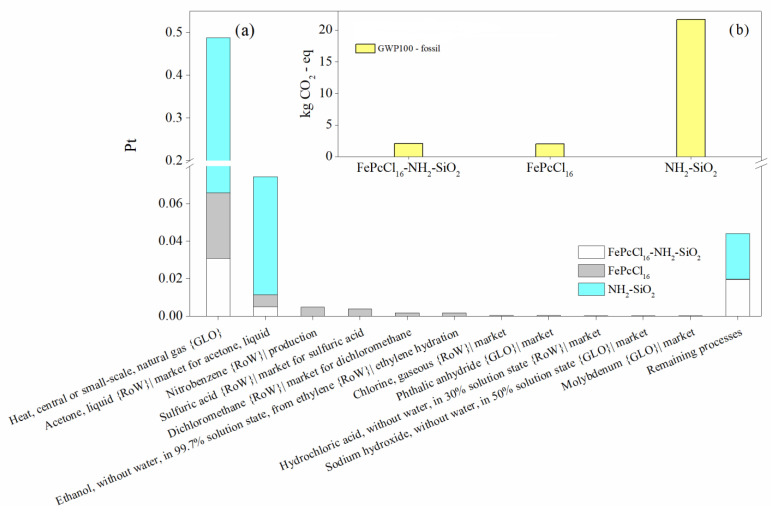
Environmental impact assessment in the production of the catalyst FePcCl_16_-NH_2_-SiO_2_, support and active phase by (**a**) process contribution and (**b**) the global warming potentials. Assessment methods: ReCiPe 2016 Endpoint (H) V1.06/World (2010) H/A; IPCC 2021 GWP100 v1.0.

**Figure 8 molecules-27-05479-f008:**
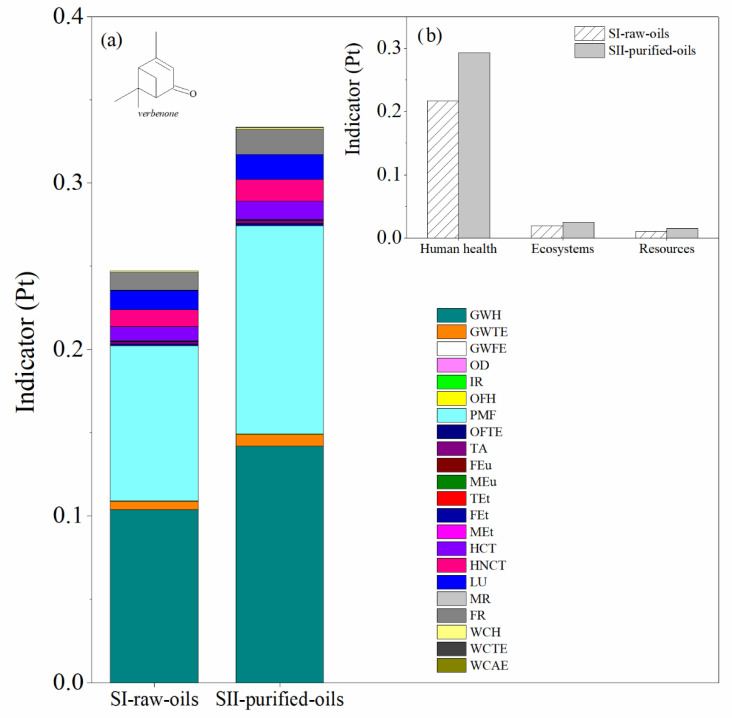
Comparison of the environmental impacts in the SI-raw-oils and SII-purified-oils scenarios for the verbenone process production. (**a**) Impact categories and (**b**) damage categories. Assessment method: ReCiPe 2016 Endpoint (H) V1.04/World (2010) H/A.

**Figure 9 molecules-27-05479-f009:**
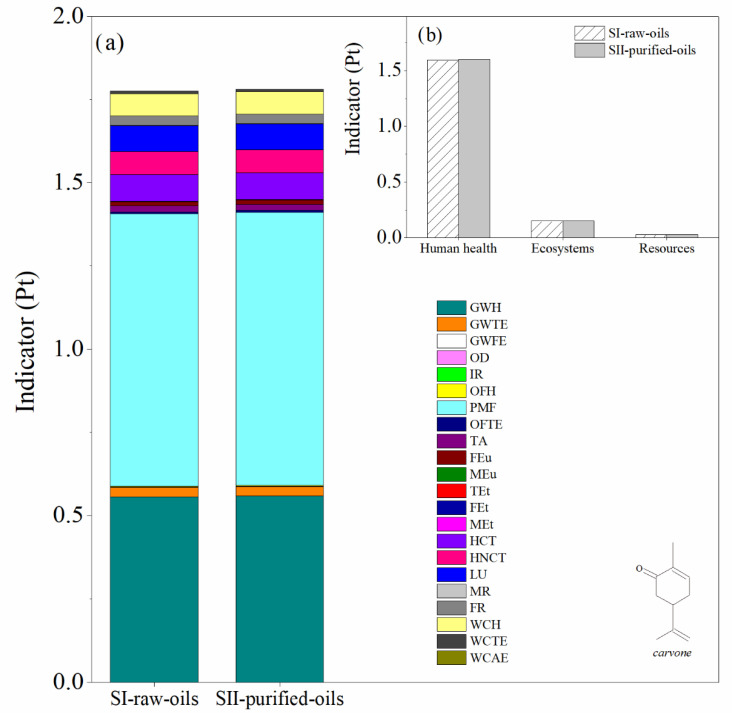
Comparison of the environmental impacts in the SI-raw-oils and SII-purified oils scenarios for the carvone process production. (**a**) Impact categories and (**b**) damage categories. Assessment method: ReCiPe 2016 Endpoint (H) V1.04/World (2010) H/A.

**Figure 10 molecules-27-05479-f010:**
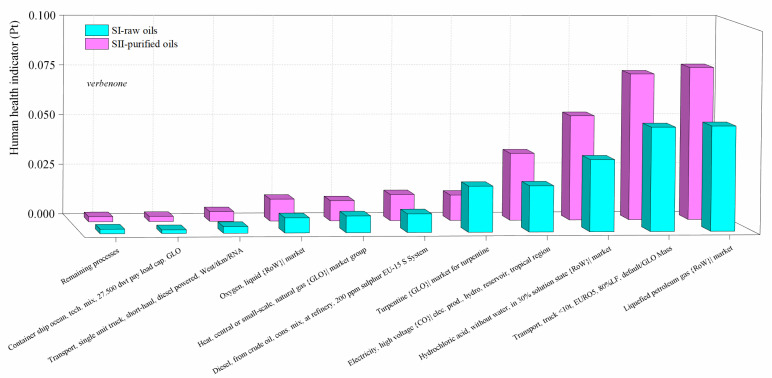
Comparison of the environmental impacts in the SI-raw-oils and SII-purified-oils scenarios for the verbenone process production by process contribution on the human health damage category. Assessment method: ReCiPe 2016 Endpoint (H) V1.04/World (2010) H/A.

**Figure 11 molecules-27-05479-f011:**
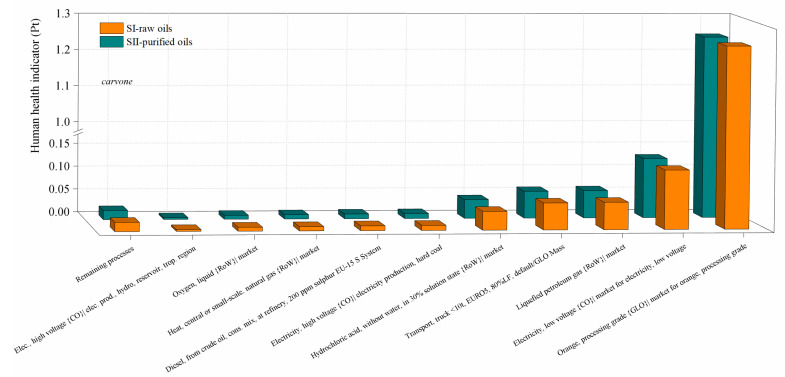
Comparison of the environmental impacts in the SI-raw-oils and SII-purified oils scenarios for the carvone process production by process contribution on the human health damage category. Assessment method: ReCiPe 2016 Endpoint (H) V1.04/World (2010) H/A.

**Figure 12 molecules-27-05479-f012:**
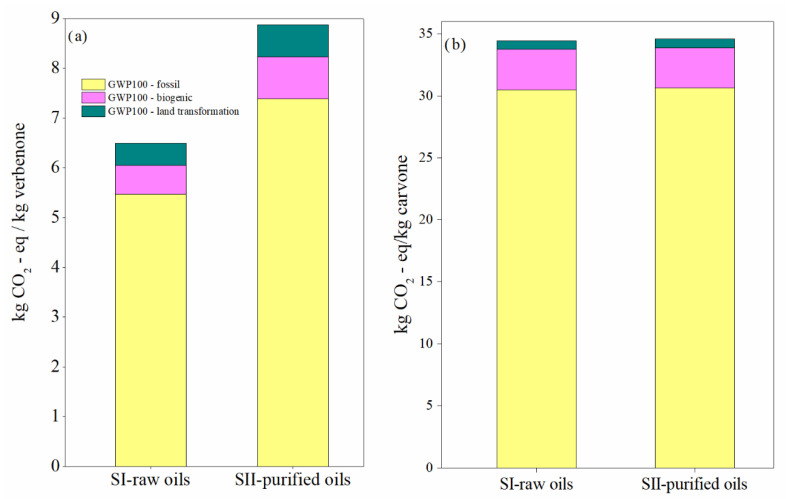
Global warming potential in the SI-raw-oils and SII-purified-oils scenarios for the (**a**) verbenone and (**b**) the carvone process production. Assessment method: IPCC 2021 GWP100 v1.0.

**Figure 13 molecules-27-05479-f013:**
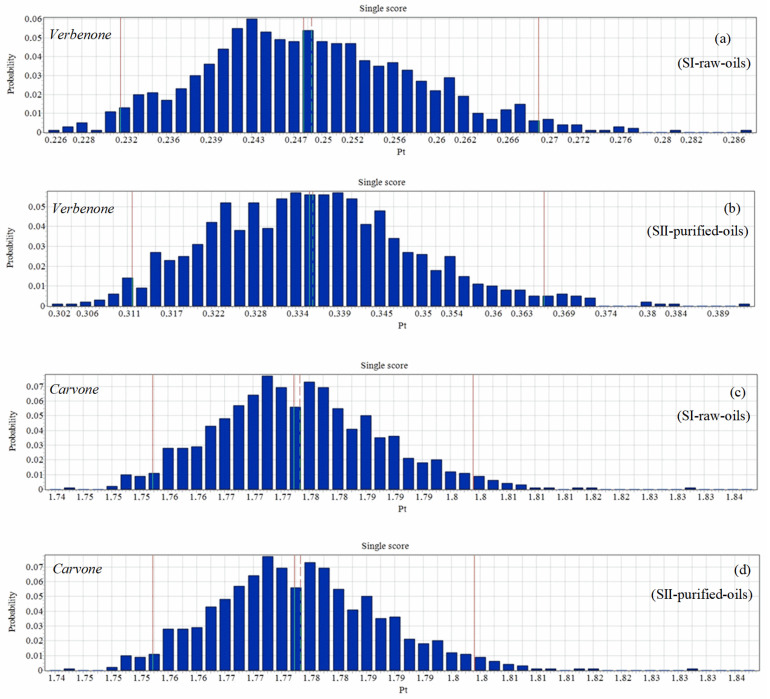
Uncertainty analysis of the process for the production of verbenone (**a**,**b**) and carvone (**c**,**d**). Assessment methods: ReCiPe 2016 Endpoint (H) V1.06/World (2010) H/A; number of bins: 50, visible interval: 99.9%, confidence interval: 95%. Functional unit: 1 kg of ketone/batch.

**Figure 14 molecules-27-05479-f014:**
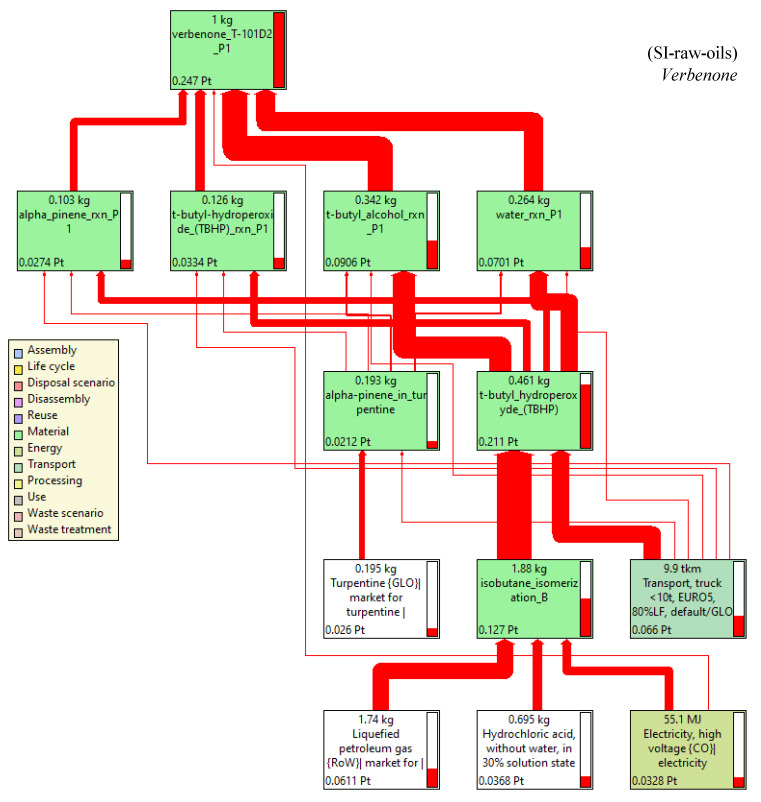
Network diagram by single score of the process for verbenone production in the SI-raw-oils scenario. Assessment method: ReCiPe 2016 Endpoint (H) V1.06/World (2010) H/A. Functional unit: 1 kg of verbenone/batch, Node cut-off: 8%, cumulative indicator as percentage.

**Table 1 molecules-27-05479-t001:** Design parameters of the proposed scenarios in the conceptual process for the production of verbenone and carvone.

	Verbenone		Carvone	
	SI-Raw-Oils	SII-Purified-Oils	SI-Raw-Oils	SII-Purified-Oils
R-101				
Initial mass, kg	1665.60	1627.85	1624.90	1615.80
Ketone mass produced, kg	27.23	27.63	13.03	10.52
Molar percentage ketone, %	0.539	0.546	0.242	0.199
Heat duty, kW	−316.8	−326.9	−530.6	−483.8
*T*, K	318	318	318	318
*P*, kPa	85	85	85	85
Time, h	6	6	6	6
F-101				
Ketone mass recovered, kg	26.96	27.36	12.90	10.41
D-101				
Molar percentage ketone concentrated, %	22.6	44.84	16.58	2.03
Reboiler heat duty, kW	363	350	383	383
Time, h	2	2	2	2
T-101				
Initial mass, kg	119.87	59.56	70.98	179.31
Product mass, kg	41.78	26.53		
Ketone purity, %	59.7	83.9	36.5	32.6
Reboiler heat duty, kW	26.8	9.6	8.3	46.0
Ketone capacity at 3 batches in R-101 per day, ton/y	46.01	29.21	36.44	29.53

**Table 2 molecules-27-05479-t002:** Impact categories selected for the analysis.

Impact Category	Abbreviation
Global warming, Human health	GWH
Global warming, Terrestrial ecotoxicity	GWTE
Global warming, Freshwater ecotoxicity	GWFE
Stratospheric Ozone Depletion	OD
Ionizing radiation	IR
Ozone formation, Human health	OFH
Fine particulate matter formation	PMF
Ozone Formation, Terrestrial Ecosystem	OFTE
Terrestrial Acidification	TA
Freshwater Eutrophication	FEu
Marine Eutrophication	MEu
Terrestrial Ecotoxicity	TEt
Freshwater ecotoxicity	FEt
Marine ecotoxicity	MEt
Humam Carcinogenic Toxicity	HCT
Human Non-carcinogenic Toxicity	HNCT
Land Use	LU
Mineral Resource Scarcity	MR
Fossil Resource Scarcity	FR
Water Consumption, Human health	WCH
Water Consumption, Terrestrial ecosystem	WCTE
Water Consumption, Aquatic ecosystem	WCAE

**Table 3 molecules-27-05479-t003:** Statistical parameters (by score) of the uncertainty analysis of the processes. Assessment methods: ReCiPe 2016 Endpoint (H) V1.06/World (2010) H/A; number of bins: 50, visible interval: 99.9%, confidence interval: 95%. Functional unit: 1 kg of ketone/batch.

Ketone Process	Verbenone		Carvone	
Scenario/Statisticals of the Score	SI-Raw-Oils	SII-Purified-Oils	SI-Raw-Oils	SII-Purified-Oils
Mean	0.249	0.335	1.78	1.78
Median	0.248	0.334	1.78	1.78
SD	0.00973	0.0138	0.0116	0.0123
CV	3.91%	4.11%	0.654%	0.691%
2.5%	0.23	0.312	1.76	1.76
97.5%	0.27	0.367	1.8	1.81
SEM (standard error of mean)	0.000308	0.000436	0.000367	0.00039

**Table 4 molecules-27-05479-t004:** Sensitivity analysis of transportation and electricity in the process for verbenone production in the SI-raw-oils scenario. Assessment method: ReCiPe 2016 Endpoint (H) V1.06/World (2010) H/A. Functional unit: 1 kg of verbenone/batch, cumulated indicator as percentage.

Transport, Truck < 10 t, Euro TYPE, 80%LF, Default/GLO Mass Sensitive Analysis			
Activity/Cumulated Indicator (%)	EURO1	EURO3	EURO5
*t*-butyl_hydroperoxyde_(TBHP)	85.8	85.7	85.2
isobutane_isomerization_B	49.1	49.3	51.1
Transport, truck < 10 t, EURO TYPE, 80%LF, default/GLO Mass	27.9	27.7	25.1
*t*-butyl_alcohol_rxn_P1	32.1	32.1	31.9
water_rxn_P1	24.8	24.8	24.7
alpha_pinene_rxn_P1	9.69	9.68	9.62
**Electricity, High Voltage {CO}| Electricity Production, TYPE**			
**Activity/Cumulated Indicator (%)**	**Natural Gas**	**Hydro**	
isobutane_isomerization_B	69.8	51.2	
*t*-butyl_hydroperoxyde_(TBHP)	69.8	51.2	
Liquified petroleum gas {RoW}|market for|Cut-off, S	14.1	23.9	
Hydrochloric acid, without water, in 30% solution state {RoW}|market for|Cut-off, S	8.52	14.4	
Electricity, high voltage {CO}| electricity production, TYPE	47.2	12.8	

## Data Availability

Data contained within the article and Appendix A are available on request from the authors.

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
