# Peer review of "Comparison by Life-Cycle Assessment of Alternative Processes for Carvone and Verbenone Production"

_molecules, 2022, doi:10.3390/molecules27175479_

Round 1

Reviewer 1 Report

I.                 General Overview

In this study, the environmental impacts in the conceptual process to produce verbenone and carvone were evaluated by ReCiPe and IPCC methods implemented on SimaPro. The topic is interesting and has significance. However, this manuscript is still crude and some substantial questions should be addressed.

II.               Specific Comments

1.      The abstract lacks a summary of the research status which cannot highlight the academic contribution and innovation of this research.

2.      The introduction from line 26-59 describes too much information about organic synthesis to produce carvone and verbenone. It is recommended that some general information should be deleted to highlight the significance of this research.

3.      Line 60-62: “the approaches to the sustainable (environmental, economic, and social) assessment to produce these compounds are not available” Why this study did a comparison LCA of for carvone and verbenone production? The practical significance of this research should be highlight rather than just lacking of assessment.

4.      Line 15-21: The results in abstract are apparently vague and general. More in-depth and profound analysis results should be given.

5.      The detailed LCI results for the batch production of verbenone and carvone should be listed in your manuscript. Some figures and tables that are too technical can be deleted in manuscript and included in supplementary materials.

6.      Line 28-29: “changing the production route of 3,4-ethylenedioxythiophene (a precursor of electrode backbones) strongly influenced the results.” What do you mean of “changing the production route”? The expression of results is too noncommittal. More detailed explanation should be given.

7.      What “mPt” does means in figure 5? You should give the full name to make it more easily understood.

8.      You should use some generic abbreviation for the impact categories to make the figures cleaner and elegant.

9.      There are too many graphs and tables in the manuscript. It is recommended that some of them should be deleted or combined to display the most important results.

10.  The compiling of background inventory in this study was based on secondary data. It is very important to ensure the data quality. So it's highly recommended to assess the data quality of these secondary data according to the reference:

Salemdeeb, R., et al. "A pragmatic and industry-oriented framework for data quality assessment of environmental footprint tools." Resources, Environment and Sustainability (2021).

11.  Most of the references in the manuscript were published before 2018, should be followed. It's recommended to add more up-of-data literatures as follows:

Kotaro Kawajiri, et al. Life cycle assessment of critical material substitution: Indium tin oxide and aluminum zinc oxide in transparent electrodes, Resources, Environment and Sustainability, 2022.

12.  The originality of the paper needs to be further clarified.

13.  The figures and tables are too rough which need to be further improved. The present form does not have sufficient results to justify the novelty of a high quality journal paper.

14.  The presentation of figures is not formal, such as the lack of vertical coordinates. Some less important figures and tables should be put in supplement materials.

15.  The reference of 40 is Chinese which should be modified to English.

Reviewer 2 Report

Dear authors 

The article has an interesting topic in the field. The presentation seems good enough to be considered in the journal. There are some points that you may need to consider more in detail. Please consider all the comments below: 

1. There are some errors and mistakes in the reference list. The reference number 40 is not acceptable in this form. 

2. The conclusion part seems too short. More discussion and managerial implications can be added to present your study more effectively. 

3. The introduction part seems too long. There is no literature review, which is why your introduction became that long. I highly suggest you add a sub-section somewhere to discuss your contributions in more detail. In this form looks harder for readers to find out promptly. 

4. Also, please add a short introduction for section number 2. Section 2 also seems too long. With an introduction, our readers can understand the procedures better and faster. 

Reviewer 3 Report

This paper is interesting and considers a factual question. Overall, it is well structured, a lot of important information being shown. In fact, data as the one presented in figure 2 and 4, and Table A2 is very relevant and self-explanatory. Nevertheless, there are some points where extra efforts are needed to better convey the message of the work, namely:

1- Please detail a bit more why the graphic bars in Figure 9 are so similar.

2 - The same for figure 12, graphic b.

3- What was the grid mix used? Would the results differ too much in the case of a grid mix composed of a higher share of renewable energy?

4- Was any type of allocation performed?

5- The authors do not specify whether this is an attributional or consequential analysis.

6- It is recommended that the authors perform a sensitivity analysis, to verify the robustness of the assessment.

Round 2

Reviewer 2 Report

Dear authors 

The article seems much better in its revised form. The article can be accepted in its current form.